

# Impact of old age on resectable colorectal cancer outcomes

Jianfei Fu[1,*], Hang Ruan[2,*], Hongjuan Zheng[1], Cheng Cai[2],
Shishi Zhou[1], Qinghua Wang[1], Wenbin Chen[3], Wei Fu[4] and Jinlin Du[2]

[1] Department of Medical Oncology, Jinhua Hospital, Zhejiang University School of Medicine, Jinhua, Zhejiang, P.R. China
[2] Department of Colorectal Surgery, Jinhua Hospital, Zhejiang University School of Medicine, Jinhua, Zhejiang, P.R. China
[3] Department of Colorectal Surgery, The First Affiliated Hospital, Zhejiang University School of Medicine, Hangzhou, Zhejiang, P.R. China
[4] Division of Oncology, Johns Hopkins University School of Medicine, Baltimore, MD, USA
* These authors contributed equally to this work.

Corresponding author
Jinlin Du, djl9090@163.com

## ABSTRACT

**Objective:** This study was performed to identify a reasonable cutoff age for defining older patients with colorectal cancer (CRC) and to examine whether old age was related with increased colorectal cancer-specific death (CSD) and poor colorectal cancer-specific survival (CSS).

**Methods:** A total of 76,858 eligible patients from the surveillance, epidemiology, and end results (SEER) database were included in this study. The Cox proportional hazard regression model and the Chow test were used to determine a suitable cutoff age for defining the older group. Furthermore, a propensity score matching analysis was performed to adjust for heterogeneity between groups. A competing risk regression model was used to explore the impact of age on CSD and non-colorectal cancer-specific death (non-CSD). Kaplan–Meier survival curves were plotted to compare CSS between groups. Also, a Cox regression model was used to validate the results. External validation was performed on data from 1998 to 2003 retrieved from the SEER database.

**Results:** Based on a cutoff age of 70 years, the examined cohort of patients was classified into a younger group ($n = 51,915$, <70 years of old) and an older group ($n = 24,943$, ≥70 years of old). Compared with younger patients, older patients were more likely to have fewer lymph nodes sampled and were less likely to receive chemotherapy and radiotherapy. When adjusted for other covariates, age-dependent differences of 5-year CSD and 5-year non-CSD were significant in the younger and older groups (15.84% and 22.42%, $P < 0.001$; 5.21% and 14.21%, $P < 0.001$). Also an age of ≥70 years remained associated with worse CSS comparing with younger group (subdistribution hazard ratio, 1.51 95% confidence interval (CI) [1.45–1.57], $P < 0.001$). The Cox regression model as a sensitivity analysis had a similar result. External validation also supported an age of 70 years as a suitable cutoff, and this older group was associated with having reduced CSS and increased CSD.

**Conclusions:** A total of 70 is a suitable cutoff age to define those considered as having elderly CRC. Elderly CRC was associated with not only increased non-CSD but also with increased CSD. Further research is needed to provide evidence of whether cases of elderly CRC should receive stronger treatment if possible.

## INTRODUCTION

Age-specific risk rises markedly in old age, and as mortality from heart disease and other non-cancer causes decrease, this leaves the elderly population at high risk for developing bowel cancer (*Papamichael et al., 2009*). The increasing incidence of colorectal cancer (CRC) in the segment of the population >70 years of age necessitates an examination of what type of treatment is most appropriate for these patients with CRC. Approximately 60% of CRC patients are >70 years of age at the time of diagnosis, and 43% are >75 years of age (*Papamichael et al., 2009*). It remains controversial whether so-called "elderly CRC" exhibits a differential prognosis compared to younger CRC and whether age (i.e., old) is an independent prognostic factor. There is now a general consensus that those in the old population possess a high frequency of frailty and comorbidities, exhibiting increased mortality from other causes among those with CRC. However, it remains unknown whether old patients have an increased incidence of colorectal cancer-specific death (CSD). A previous report demonstrated that older patients with CRC who survived the first year after surgery exhibited the same overall cancer-related survival as did younger patients (*Dekker et al., 2011*). Of note, chronological age is distinct from biological age (*Odden et al., 2012*; *Robinson et al., 2009*). The definition of "elderly" differs, being given as anything between >65 years and >80 years in different studies (*Seymour, 2004*; *Twelves et al., 2005*). As such, it is difficult to know whether 70 years is a reasonable cutoff age to safely extrapolate these results or whether the decision should depend on the physical and functional status of the patient rather than just on chronological age. Unlike younger patients with CRC, wherein more reliable evidence-based guideline based on clinical trials are available, older patients are still in need of increased evidence to guide clinical practices due to most trials excluding older patients with CRC. We used big data to explore the impact of old age on CRC prognoses to help guide clinical practice in the treatment of elderly CRC patients.

## MATERIALS AND METHODS

### Data

Data on colon cancer records from 18 cancer registries in the National Cancer Institute's surveillance, epidemiology and end results (SEER) cancer database based on the November 2015 submission were collected. SEER.Stat software was utilized to identify patients with resectable stage I–III CRC, and information regarding chemotherapy was obtained by submitting a special data request to the SEER program.

Cohort inclusion criteria were as follows: (1) Years of diagnosis from 2004 to 2011. (2) Patients diagnosed with stage I–III CRC. (3) Patients received surgery (Surgery code 30–80). (4) Histological type ICD-O-3 was limited to 8140/3, 8480/3, 8481/3, and 8490/3. Exclusion criteria were as follows: (1) Patients lacking documentation of age at diagnosis, gender, race, marital status, differentiated grade, and classification

T. (2) Patients younger than 20 years or older than 80 years. (3) Patients with multiple primary tumors. (4) Patients who survived less than one month from diagnosis. (5) Cause of death was unknown. (6) Number of lymph nodes (nLN) sampled was unknown. (7) The number of positive lymph nodes was unknown.

## Variables declaration

Patients were classified into younger (<70 years old) and older (≥70 years old) groups based on the defined cutoff age of 70 years. Race was divided into white, black and other. Marital status was categorized as married, single (never married, unmarried, or domestic partner) or divorced (separated, widowed, and divorced). Tumor location was grouped into left CRC or right CRC. Left CRC includes the rectum, rectosigmoid junction, sigmoid colon, descending colon, and splenic flexure. Right CRC includes the transverse colon, hepatic flexure, ascending colon, cecum, and appendix. Histological type was categorized as adenocarcinoma, mucinous adenocarcinoma, or ring signet cell cancer. All cases were regrouped according to the seventh American Joint Committee on Cancer TNM staging system. nLN sampled was regrouped as 0, 1–3, 4–6, 7–11, and ≥12. The variable chemotherapy was classified as chemotherapy "yes" or "no/unknown" according to the SEER program (*Noone et al., 2016*).

## Statistical analysis

The restrict cubic spline function, "RCS," with three knots was used to transform the continuous variable of age. The "rcspline.plot" function provided plots of the estimated restricted cubic spline function relating a single predictor (age) to the response for a Cox model. The Chow test method (Fstats and breakpoints in strucchange package) was used to explore a suitable cutoff value for age to define elderly CRC. Differential distribution of clinicopathological characteristics between younger and older subgroups was indicated by standardized difference (*Austin, 2009*). Propensity scores were used to balance the difference of distribution between younger and older groups on sociodemographic and clinical characteristics. Matchit package in R software was used as the nearest method with ratio 1:1. CSD and non-cancer-specific death (non-CSD) were considered the primary endpoint and were calculated by the Gray test (*Howlader et al., 2014*). The secondary endpoint was cancer-specific survival (CSS). The non-CSD referred to the dead due to other causes. When CSDs were calculated, follow up time was calculated from the date of diagnosis to the date of death from CRC. Alive were defined as censored, and the non-CSD was considered a competing event. The subdistribution hazard ratio (SHR) of variables for cause-specific death was estimated using the Fine and Gray proportional hazard model (*Fine & Gray, 1999*). As a comparison, the hazard ratio (HR) of variables was also estimated with a Cox proportional hazards model.

The cutoff of age and prognostic value of old were validated externally using a cohort from 1998 to 2003. A standard difference that was less than 0.1 indicated a negligible difference in the mean or prevalence of a covariate between groups (*Normand et al., 2001*). When a two-sided $P$-value < 0.05, the difference was considered statistically significant. All statistical analyses were performed using R software (*R Core Team, 2016*).

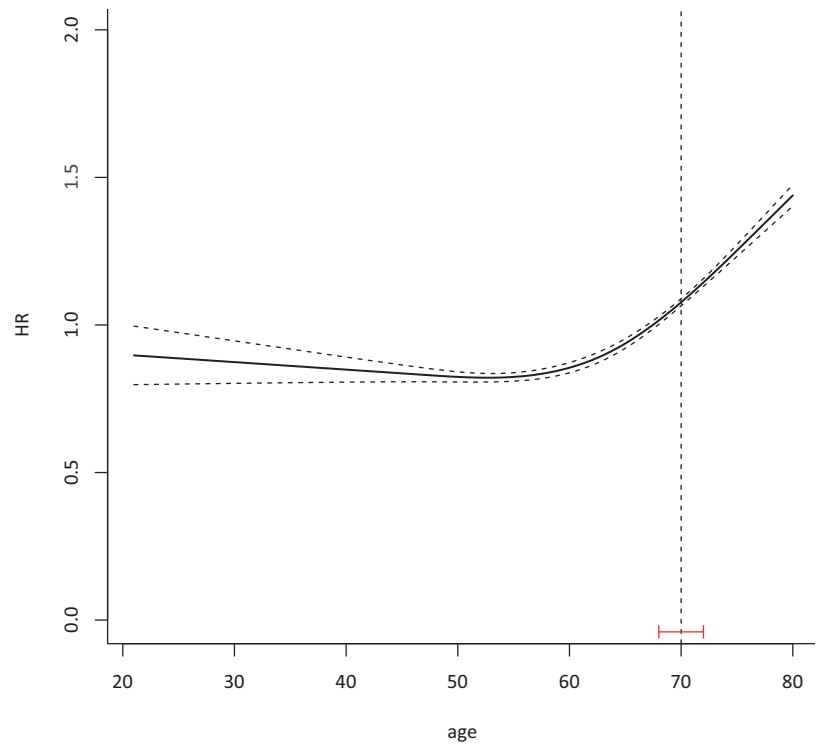

**Figure 1** **A total of 70 was identified as a suitable cutoff age to define elderly CRC with 95% CI 68–72.**
Cox proportional hazard model with continued variable of age after transformation with restrict cubic spline method was plotted to examine the relationship between age and HR of colorectal cancer-specific death 76,858 eligible patients were included in the test cohort with dummy variables for each age to attain the HR of every age. The Chow test was used to determine a suitable cutoff age.

## RESULTS

### Baseline characteristics and identification of the old age cutoff value

A total of 76,858 eligible patients were included in this cohort. The endpoint date for follow-up was November 2013, and the median follow-up time was 55.0 months (range 1.0–119.0 months).

The median age of patients was 64 years (IQR 20–80 years). With age as a continuous variable, the HR of CSS was 1.54 (95% CI 1.39–1.69, $P < 0.001$). The HR of CSS slowly increased before 70 years of age, and then increased significantly after 70 years according to the Cox model (Fig. 1).

### Clinicopathological features of old colorectal cancer patients

Based on the cutoff age of 70 years, the test cohort of patients was classified into the following two groups: younger group ($n = 51,915$, <70 years of old) or older group ($n = 24,943$, ≥70 years of old). Older patients exhibited a high frequency of male, Caucasians, right CRC, mucinous carcinoma, more poorly differentiated grade and earlier stage. Detailed clinicopathological characteristics of the chemotherapy subgroups are presented in Table 1. Compared with younger patients, older patients were more

**Table 1  The characteristics of 76,858 colorectal cancer patients in the younger and older groups.**

| Characteristics | Younger (<70 years) Before PSM N (%) | Older (≥70 years) Before PSM N (%) | SD | Younger (≤70 years) After PSM N (%) | Older (≥70 years) After PSM N (%) | SD |
|---|---|---|---|---|---|---|
| **Gender** | | | | | | |
| Female | 22,748 (43.82) | 12,925 (51.82) | 0.57 | 11,648 (49.41) | 11,924 (50.59) | 0.02 |
| Male | 29,167 (56.18) | 12,018 (48.18) | 0.92 | 12,160 (50.57) | 11,884 (49.43) | 0.02 |
| **Marital status** | | | | | | |
| Married | 33,210 (63.97) | 14,227 (57.04) | 0.87 | 14,646 (50.78) | 14,197 (49.22) | 0.03 |
| Single | 9,416 (18.14) | 2,314 (9.28) | 1.52 | 2,370 (50.60) | 2,314 (49.40) | 0.02 |
| Divorced | 9,289 (17.89) | 8,402 (33.68) | 0.10 | 6,792 (48.21) | 7,297 (51.79) | 0.07 |
| **Race** | | | | | | |
| White | 39,981 (77.01) | 20,482 (82.12) | 0.68 | 19,317 (49.92) | 19,379 (50.08) | <0.01 |
| Black | 6,768 (13.04) | 2,289 (9.18) | 1.14 | 2,339 (50.55) | 2,288 (49.45) | 0.02 |
| Others | 5,166 (9.95) | 2,172 (8.71) | 0.89 | 2,152 (50.13) | 2,141 (49.87) | 0.01 |
| **Location** | | | | | | |
| Left | 33,719 (64.95) | 12,307 (49.34) | 1.05 | 12,470 (50.69) | 12,131 (49.31) | 0.03 |
| Right | 18,196 (35.05) | 12,636 (50.66) | 0.37 | 11,338 (49.26) | 11,677 (50.74) | 0.03 |
| **Histological type** | | | | | | |
| Adenocarcinoma | 46,246 (89.08) | 21,952 (88.01) | 0.76 | 21,056 (50.23) | 20,862 (49.77) | 0.01 |
| Mucinous adenocarcinoma | 5,181 (9.98) | 2,778 (11.14) | 0.63 | 2,538 (48.14) | 2,734 (51.86) | 0.07 |
| Signet ring cell cancer | 488 (0.94) | 213 (0.85) | 0.85 | 214 (50.23) | 212 (49.77) | 0.01 |
| **Differentiated grade** | | | | | | |
| Well | 4,225 (8.14) | 1,983 (7.95) | 0.77 | 1,952 (50.03) | 1,950 (49.97) | <0.01 |
| Moderate | 39,154 (75.42) | 18,409 (73.80) | 0.77 | 17,677 (50.41) | 17,391 (49.59) | 0.02 |
| Poor | 8,536 (16.44) | 4,551 (18.25) | 0.64 | 4,179 (48.33) | 4,467 (51.67) | 0.07 |
| **T-classification[a]** | | | | | | |
| T1 | 4,594 (8.85) | 2,179 (8.74) | 0.76 | 2,178 (50.09) | 2,170 (49.91) | <0.01 |
| T2 | 8,546 (16.46) | 4,580 (18.36) | 0.63 | 4,429 (49.92) | 4,443 (50.08) | <0.01 |
| T3 | 32,608 (62.81) | 15,320 (61.42) | 0.77 | 14,518 (50.25) | 14,376 (49.75) | 0.01 |
| T4 | 6,167 (11.88) | 2,864 (11.48) | 0.79 | 2,683 (48.76) | 2,819 (51.24) | 0.05 |
| **N-classification[a]** | | | | | | |
| N0 | 29,285 (56.41) | 15,866 (63.61) | 0.62 | 15,072 (50.35) | 14,863 (49.65) | 0.01 |
| N1 | 14,542 (28.01) | 6,083 (24.39) | 0.90 | 5,764 (49.11) | 5,972 (50.89) | 0.04 |
| N2 | 8,088 (15.58) | 2,994 (12.00) | 1.04 | 2,972 (49.99) | 2,973 (50.01) | <0.01 |
| **nLN** | | | | | | |
| 0 | 2,733 (5.26) | 1,343 (5.38) | 0.73 | 1,324 (50.06) | 1,321 (49.94) | <0.01 |
| 1–2 | 872 (1.68) | 456 (1.83) | 0.66 | 407 (47.22) | 455 (52.78) | 0.11 |
| 3–5 | 2,518 (4.85) | 1,422 (5.70) | 0.58 | 1,314 (50.00) | 1,314 (50.00) | <0.01 |
| 6–11 | 9,808 (18.89) | 5,494 (22.03) | 0.59 | 5,013 (49.18) | 5,180 (50.82) | 0.03 |
| ≥12 | 35,984 (69.31) | 16,228 (65.06) | 0.82 | 15,750 (50.34) | 15,538 (49.66) | 0.01 |
| **Stage[a]** | | | | | | |
| I | 10,334 (19.91) | 5,741 (23.02) | 0.60 | 5,647 (50.23) | 5,596 (49.77) | 0.01 |
| II | 18,951 (36.50) | 10,125 (40.59) | 0.64 | 9,425 (50.42) | 9,267 (49.58) | 0.02 |

(Continued)

| Characteristics | Younger (<70 years) Before PSM N (%) | Older (≥70 years) Before PSM N (%) | SD | Younger (≤70 years) After PSM N (%) | Older (≥70 years) After PSM N (%) | SD |
|---|---|---|---|---|---|---|
| III | 22,630 (43.59) | 9,077 (36.39) | 0.95 | 8,736 (49.41) | 8,945 (50.59) | 0.02 |
| CT | | | | | | |
| No | 24,094 (46.41) | 16,990 (68.12) | 0.35 | 15,580 (49.56) | 15,856 (50.44) | 0.02 |
| Yes | 27,821 (53.59) | 7,953 (31.88) | 1.34 | 8,228 (50.85) | 7,952 (49.15) | 0.03 |
| RT | | | | | | |
| No | 40,769 (78.53) | 22,233 (89.14) | 0.62 | 20,949 (49.82) | 21,098 (50.18) | 0.01 |
| Yes | 11,146 (21.47) | 2,710 (10.86) | 1.53 | 2,859 (51.34) | 2,710 (48.66) | 0.05 |

**Notes:**
[a] Stage TNM, T, N-classification to seventh edition of AJCC staging system.
All statistical tests were two-sided.
Abbreviations: PSM, propensity score matching; SD, standardized difference; nLN, number of lymph nodes; CT, chemotherapy treatment; RT, radiotherapy treatment.
Left includes rectum, rectosigmoid junction, sigmoid colon, descending colon, and splenic flexure.
Right includes transverse colon, hepatic flexure, ascending colon, cecum, and appendix.

likely to have fewer lymph nodes sampled and were less likely to receive chemotherapy and radiotherapy.

The sample size of patients in the older group was obviously fewer than in the younger group, and these groups had different clinical characteristics, so a method for propensity score matching (PSM) was used to balance differences in baseline characteristics and generate a corrected test cohort. Most covariates were well balanced between younger and older groups in the corrected test cohort (Table 1).

## Competing risk regression model was used to explore the impact of age on CSD and non-CSD

A total of 14,425 (18.77%) and 6,982 (9.08%) patients died of CSD and non-CSD, respectively. The corrected test cohort after PSM, a total of 47,616 patients analyzed, showed that 9,273 (19.47%) and 5,514 (11.58%) patients died of CSD and non-CSD, respectively. The 5-year CSD in the younger and older groups were 15.84% and 22.42%, respectively, and were significantly different ($P < 0.001$). The 5-year non-CSD in the younger and older groups were 5.21% and 14.21%, respectively, and were also significantly different ($P < 0.001$) (Fig. 2). Univariate and multivariate analyses demonstrated that old age was associated with CSD (Table 2).

## Subgroup analysis by characteristics in raw data

Subgroup analyses were performed based on gender, race, differentiated type, pathological type, T and N classification, nLN, TNM stage, chemotherapy, and radiotherapy in the raw data. In all subsets (except for ring signet cell cancer), patients in the older group exhibited poorer prognosis compared with those in the younger group (Fig. 3). Furthermore, we performed an interaction analysis using Cox model between the therapy and age in stage III patients in raw data (Fig. S2). The results showed that the older group benefited more from chemotherapy than younger group

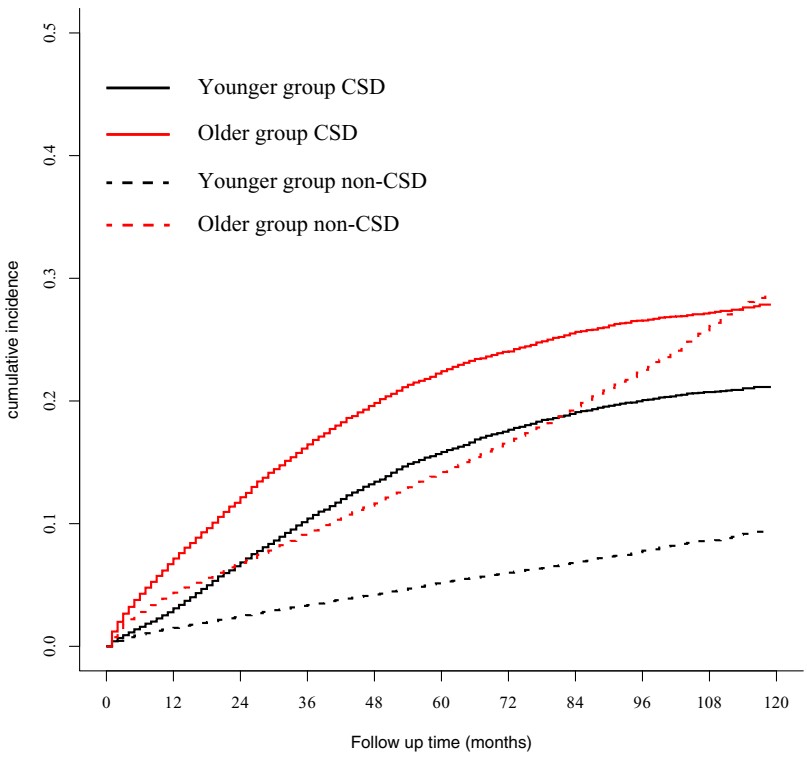

**Figure 2 Gray method showed cumulative incidence curves of CSD and non-CSD in younger and older groups.** CSD, cancer-specific death; non-CSD, non-cancer-specific death.

(*P*-value = 0.001). However, the interaction between radiotherapy and old was not significant (*P*-value = 0.328).

## Univariate and multivariate Cox regression analysis

A Kaplan–Meier survival curve for CSS is presented in Fig. 4. Univariate analysis showed a HR for CSS of older patients of 1.57 (95% CI 1.51–1.64, *P* < 0.001, reference to younger group) in the corrected cohort. Multivariate analysis showed that the HR for CSS in older patients was 1.64 (95% CI 1.57–1.70, *P* < 0.001, reference to younger group) (for detailed data see Table 3; raw data are shown in Table S1).

## External validation

In the validated cohort, 66,946 patients from 1998 to 2003 were retrieved from the SEER dataset. The corrected validated cohort after PSM was used to validate the above results. The relationship between age and colorectal CSD for Cox model presented a single arm "U" shape (Fig. S1). Seventy was still the cutoff age in the validated cohort. Detailed clinicopathological characteristics are presented in Table S2. In the corrected validated cohort, there were few differences in the distribution of different clinicopathological factors between older and younger groups (see Table S2). Univariate and multivariate analysis showed that old age is related with poor CSS (based on the Cox model) and increased CSD (based on the competing risk model) (see Table S3).

**Table 2 Univariate and multivariate analysis of colorectal cancer-specific death of 47,616 patients after PSM.**

| Risk factors | Univariate analysis | | Multivariate analysis | |
|---|---|---|---|---|
| | SHR (95%CI) | P[b] | SHR (95%CI) | P[b] |
| Age | | | | |
| <70 | 1 | | 1 | |
| ≥70 | 1.47 (1.41–1.53) | <0.001 | 1.50 (1.44–1.56) | <0.001 |
| Gender | | | | |
| Female | 1 | | 1 | |
| Male | 1.06 (1.02–1.10) | 0.005 | 1.11 (1.06–1.16) | <0.001 |
| Marital status | | | | |
| Married | 1 | | 1 | |
| Unmarried | 1.48 (1.39–1.58) | <0.001 | 1.27 (1.19–1.36) | <0.001 |
| Divorced | 1.35 (1.30–1.42) | <0.001 | 1.25 (1.19–1.31) | <0.001 |
| Race | | | | |
| White | 1 | | 1 | |
| Black | 1.31 (1.23–1.40) | <0.001 | 1.26 (1.18–1.35) | <0.001 |
| Other | 0.95 (0.88–1.02) | 0.176 | 0.90 (0.84–0.97) | 0.008 |
| Location | | | | |
| Left | 1 | | 1 | |
| Right | 0.93 (0.89–0.97) | <0.001 | 1.01 (0.96–1.06) | 0.759 |
| Histology | | | | |
| Adenocarcinoma | 1 | | 1 | |
| Mucinous adenocarcinoma | 1.14 (1.07–1.21) | <0.001 | 1.08 (1.01–1.15) | 0.018 |
| Signet ring cell carcinoma | 3.05 (2.62–3.54) | <0.001 | 1.48 (1.26–1.74) | <0.001 |
| Differentiated grade | | | | |
| Grade I | 1 | | 1 | |
| Grade II | 1.43 (1.31–1.56) | <0.001 | 1.27 (1.16–1.39) | <0.001 |
| Grade III | 2.45 (2.23–2.69) | <0.001 | 1.59 (1.45–1.76) | <0.001 |
| T-classification[a] | | | | |
| T1 | 1 | | 1 | |
| T2 | 0.73 (0.65–0.83) | <0.001 | 0.96 (0.85–1.08) | 0.472 |
| T3 | 2.15 (1.95–2.37) | <0.001 | 2.19 (1.98–2.43) | <0.001 |
| T4 | 5.46 (4.92–6.05) | <0.001 | 4.77 (4.27–5.33) | <0.001 |
| N-classification[a] | | | | |
| N0 | 1 | | 1 | |
| N1 | 2.33 (2.22–2.44) | <0.001 | 2.16 (2.05–2.28) | <0.001 |
| N2 | 4.48 (4.26–4.71) | <0.001 | 4.05 (3.81–4.30) | <0.001 |
| nLN | | | | |
| 0 | 1 | | 1 | |
| 0–2 | 0.49 (0.42–0.58) | <0.001 | 0.39 (0.33–0.46) | <0.001 |
| 3–5 | 0.61 (0.55–0.68) | <0.001 | 0.43 (0.39–0.48) | <0.001 |
| 6–11 | 0.56 (0.52–0.61) | <0.001 | 0.33 (0.30–0.36) | <0.001 |
| ≥12 | 0.47 (0.44–0.51) | <0.001 | 0.24 (0.22–0.26) | <0.001 |

| Risk factors | Univariate analysis | | Multivariate analysis | |
|---|---|---|---|---|
| | SHR (95%CI) | $P^b$ | SHR (95%CI) | $P^b$ |
| CT | | | | |
| No | 1 | | 1 | |
| Yes | 1.70 (1.64–1.77) | <0.001 | 0.84 (0.80–0.89) | <0.001 |
| RT | | | | |
| No | 1 | | 1 | |
| Yes | 1.35 (1.28–1.43) | <0.001 | 1.15 (1.08–1.23) | <0.001 |

**Notes:**
Left includes rectum, rectosigmoid junction, sigmoid colon, descending colon, and splenic flexure.
Right includes transverse colon, hepatic flexure, ascending colon, cecum, and appendix.
PSM, propensity score matching; CSD, cancer-specific death; nLN, number of lymph nodes; CT, chemotherapy treatment; RT, radiotherapy treatment; SHR, subdistribution hazard ratio.
[a] T-classification according to seventh AJCC staging system.
[b] P-values obtained from the $\chi2$ test. All statistical tests were two-sided.

## DISCUSSION

The statistical methods "RCS" and "Chow test" were used to determine that 70 years is a reasonable cutoff age to define elderly CRC. Elderly CRC included a high frequency of male patients, right site CRC, mucinous carcinoma, more poorly differentiated grades and earlier stages. There were fewer than 12 lymph nodes sampled and earlier stages in the older group. After eliminating the distribution difference between older and younger groups by PSM, elderly CRC had worse outcomes (CSS), and age was shown to be an independent prognostic factor, while elderly CRC was related with increased non-CSD as well as CSD. In almost all subgroups, elderly CRC exhibited worse outcomes. The external cohort validated the reasonability of 70 years as a cutoff age to define elderly CRC and further confirmed that elderly CRC exhibited worse outcomes (CSS or CSD).

In clinical practice, the optimal cutoff age is anticipated to define elderly CRC. The screening program for CRC defined 65 years as a cutoff age (*Papamichael et al., 2009*). In other previously published studies (*McCleary et al., 2013a*; *Merchant et al., 2017*; *Tournigand et al., 2012*; *Twelves et al., 2005*), 70 years was adopted as a cutoff age. Updated SEER-Medicare analysis data and three population-based data sets conducted by *Sanoff et al. (2012)* showed that only 44% of the 5,941 patients evaluated received adjuvant chemotherapy within 3 months of surgical resection for stage III CRC. In their study, 65 years was used to define elderly CRC. In clinical trials, 75 years was more frequently set as the upper limit; therefore, a more real-world data analysis adopted 75 years as their cutoff age (*Van Erning et al., 2013*). We concluded that 70 years should be adopted as a suitable cutoff age.

Once the cutoff age was defined, we found that elderly CRC exhibited a significantly different outcome than young CRC. Several studies have shown that elderly CRC has comparable outcomes compared with younger CRC. *Dekker et al.'s (2011)* study showed that if one excluded death due to operation comorbidities (mostly death occurring 1 year after an operation), there were rather similar outcomes between the younger and older groups. Late period survival was similar between older and younger subgroups

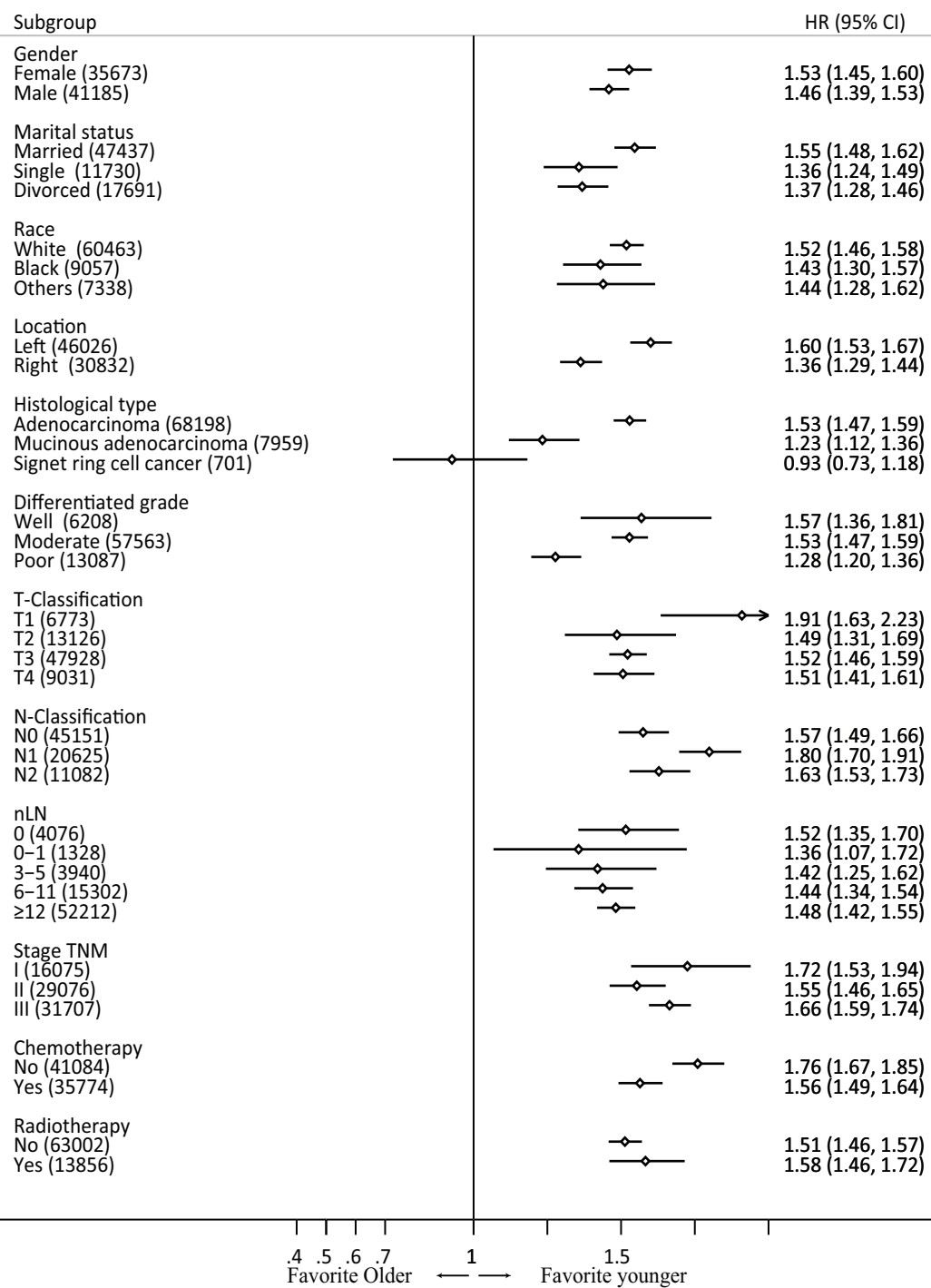

**Figure 3 Forest plot of cancer-specific survival by patient subgroup in raw data.** HR, hazard ratio.

for resectable CRC (*Dekker et al., 2011*). One Canadian study (*Merchant et al., 2017*) showed that elderly CRC did exhibit worse prognosis and was associated with a high Charlson index. In their study, they did not differentiate between CSD and non-CSD; therefore, their study did not conclude that elderly CRC was related with increased CSD.

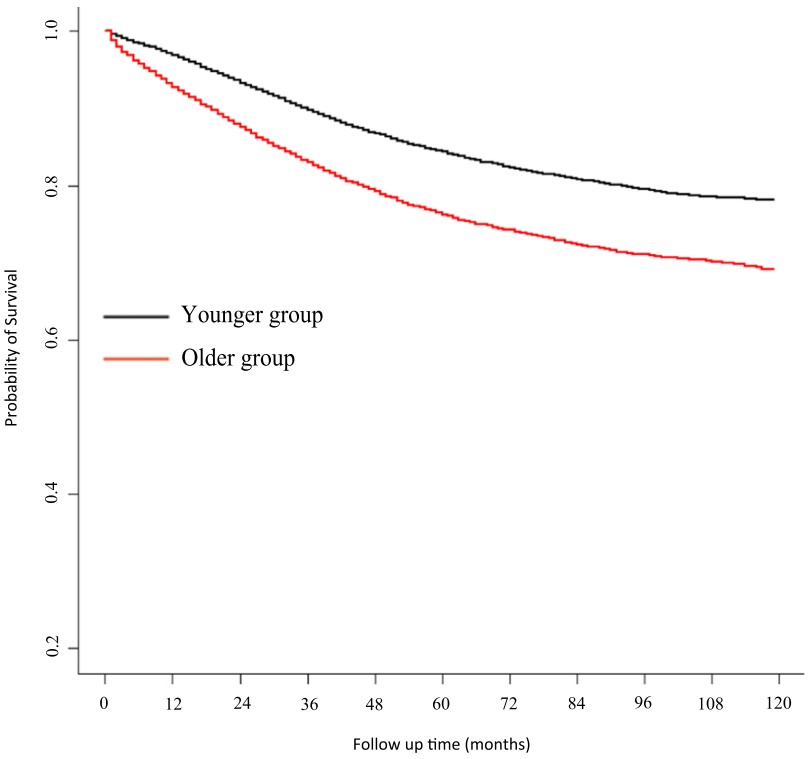

**Figure 4  Kaplan–Meier estimate of cancer-specific survival.**

In our study, we used a competing risk model to distinguish non-CSD and CSD. Furthermore, we report that elderly CRC is associated with increased CSD. In late period follow up, the Gray's cumulative events curve on CSD separated more clearly, indicating that elderly CRC is associated with worse CSD, which might be due to different register periods. Therefore, we included patients from different periods as a validation cohort. The validation cohort also confirmed that elderly CRC was not only related with increased CSD but also with increased non-CSD. Our study confirmed that elderly CRC exhibits poorer prognosis from several aspects. Except for external validation and competing risk mode, we also confirm that elderly CRC is associated with worse outcomes using PSM to balance the differential distribution.

Worse outcome of elderly CRC is multi-faceted. Fewer numbers of lymph nodes were removed from those with elderly CRC, contributing to residual tumor matter in the anticipated dissection region (mean to R1 resection) and underestimated tumor stages. Moreover, elderly CRC exhibited less capacity to endure stronger treatments, resulting in lower intensity adjuvant chemotherapy and radiotherapy. Our correlation analysis indicated that elderly CRC was related with lower incidence of chemotherapy. The outcome of less frequent chemotherapy was similar to findings in previous studies (*Abraham et al., 2013*; *Kahn et al., 2010*; *Van Erning et al., 2013*, *2014*). Finally, a CRC screening program resulted in the increased detection of precancerous lesions (such as polyps) that were treated in the old population. If the CRC diagnosis escaped screening, there was typically increased short-term carcinogenesis and more aggressive

**Table 3 Univariate and multivariate analysis of colorectal cancer-specific survival of 47,616 patients after PSM.**

| Risk factors | Univariate analysis | | Multivariate analysis | |
|---|---|---|---|---|
| | HR (95%CI) | $P^b$ | HR (95%CI) | $P^b$ |
| Age | | | | |
| <70 | 1 | | 1 | |
| ≥70 | 1.57 (1.51–1.64) | <0.001 | 1.64 (1.57–1.70) | <0.001 |
| Gender | | | | |
| Female | 1 | | 1 | |
| Male | 1.08 (1.04–1.13) | <0.001 | 1.14 (1.10–1.19) | <0.001 |
| Marital status | | | | |
| Married | 1 | | 1 | |
| Unmarried | 1.54 (1.44–1.64) | <0.001 | 1.33 (1.25–1.42) | <0.001 |
| Divorced | 1.39 (1.33–1.46) | <0.001 | 1.29 (1.23–1.35) | <0.001 |
| Race | | | | |
| White | 1 | | 1 | |
| Black | 1.34 (1.26–1.43) | <0.001 | 1.29 (1.21–1.38) | <0.001 |
| Other | 0.93 (0.86–1.00) | 0.06 | 0.87 (0.81–0.94) | <0.001 |
| Location | | | | |
| Left | 1 | | 1 | |
| Right | 0.93 (0.89–0.97) | 0.001 | 1.01 (0.96–1.06) | 0.697 |
| Histology | | | | |
| Adenocarcinoma | 1 | | 1 | |
| Mucinous adenocarcinoma | 1.13 (1.07–1.21) | <0.001 | 1.07 (1.01–1.14) | 0.033 |
| Signet ring cell carcinoma | 3.10 (2.69–3.58) | <0.001 | 1.55 (1.34–1.79) | <0.001 |
| Differentiated grade | | | | |
| Grade I | 1 | | 1 | |
| Grade II | 1.43 (1.31–1.57) | <0.001 | 1.30 (1.19–1.42) | <0.001 |
| Grade III | 2.47 (2.25–2.72) | <0.001 | 1.64 (1.49–1.81) | <0.001 |
| T-classification[a] | | | | |
| T1 | 1 | | 1 | |
| T2 | 0.73 (0.64–0.82) | <0.001 | 0.96 (0.85–1.08) | 0.488 |
| T3 | 2.15 (1.95–2.37) | <0.001 | 2.24 (2.02–2.49) | <0.001 |
| T4 | 5.57 (5.02–6.17) | <0.001 | 5.01 (4.49–5.59) | <0.001 |
| N-classification[a] | | | | |
| N0 | 1 | | 1 | |
| N1 | 2.33 (2.23–2.45) | <0.001 | 2.22 (2.10–2.34) | <0.001 |
| N2 | 4.53 (4.31–4.77) | <0.001 | 4.24 (4.00–4.49) | <0.001 |
| nLN | | | | |
| 0 | 1 | | 1 | |
| 0–2 | 0.48 (0.40–0.56) | <0.001 | 0.37 (0.31–0.43) | <0.001 |
| 3–5 | 0.58 (0.53–0.65) | <0.001 | 0.41 (0.37–0.46) | <0.001 |
| 6–11 | 0.54 (0.49–0.58) | <0.001 | 0.30 (0.28–0.33) | <0.001 |
| ≥12 | 0.44 (0.41–0.48) | <0.001 | 0.22 (0.20–0.23) | <0.001 |

| Risk factors | Univariate analysis | | Multivariate analysis | |
|---|---|---|---|---|
| | HR (95%CI) | $P^b$ | HR (95%CI) | $P^b$ |
| CT | | | | |
| No | 1 | | 1 | |
| Yes | 1.65 (1.59–1.72) | <0.001 | 0.78 (0.75–0.82) | <0.001 |
| RT | | | | |
| No | 1 | | 1 | |
| Yes | 1.32 (1.25–1.40) | <0.001 | 1.14 (1.07–1.22) | <0.001 |

Notes:
Left includes rectum, rectosigmoid junction, sigmoid colon, descending colon, and splenic flexure.
Right includes transverse colon, hepatic flexure, ascending colon, cecum, and appendix.
PSM, propensity score matching; CSD, cancer-specific death; nLN, number of lymph nodes; CT, chemotherapy treatment; RT, radiotherapy treatment; HR, hazard ratio.
[a] T-classification according to seventh AJCC staging system.
[b] P-values obtained from the $\chi2$ test. All statistical tests were two-sided.

tumor behavior. Another study showed that elderly CRC had a greater index of genetic mutations and that the incidence of BRAF mutations was higher. *Berg et al. (2010)* indicated that CIMP tumors are more common in the older population, who also have a higher rate of KRAS and BRAF mutations.

Elderly CRC with worse outcomes might require stronger treatments; however, a previous study suggested that elderly CRC does not require enhanced treatment. Many elderly patients will benefit from radical treatment approaches, but others will not, and in some cases, non-operative "palliative" management should be offered, even though the cancer is "curable." Guidelines from the International Society of Geriatric Oncology did not recommend that elderly CRC patients regularly receive adjuvant chemotherapy with limited evidence to support the benefit from such strategy (*Papamichael et al., 2009*). MOSAIC: lyses showed there to be no statistically significant benefit conferred by addition of oxaliplatin in terms of disease-free survival (DFS) or OS for older patients (70–75 years), although female patients 70–75 years of age exhibited the same oxaliplatin benefit as did younger patients (*Tournigand et al., 2012*). Interestingly, the DFS and OS benefits in patients 70–75 years were similar to those of younger patients for the first 3 years of follow-up but were lost later on due to deaths from other causes (*Andre et al., 2009*; *Tournigand et al., 2012*). NSABP-C-07: Patients ≥70 years failed to derive a statistically significant DFS or OS benefit from addition of oxaliplatin (*Tournigand et al., 2012*). Indeed, those patients receiving FLOX had poorer survival, which was attributed to toxicity. XELOXA (NO16968): the benefits observed for XELOX were maintained, although to a lesser degree in patients ≥65 and ≥70 years of age, in contrast to the results from MOSAIC and NSABP-C-07 trials (*Twelves et al., 2012*). Meta-analysis of ACCENT did not support that patients ≥70 years receive additional oxaliplatin chemotherapy (*McCleary et al., 2013a*, *2013b*). In contrast, from an analysis using real-world data, almost all showed that elderly CRC can benefit from adjuvant chemotherapy with acceptable toxicity (*Abraham et al., 2013*; *Kahn et al., 2010*; *Sanoff et al., 2012*; *Sanoff & Goldberg, 2007*; *Schrag et al., 2001*; *Van Erning et al., 2013*,

2014). One study using SEER-Medicare data indicated that elderly CRC tolerated chemotherapy well, exhibiting worse prognosis due to reduced treatment (*Sanoff et al., 2012*). More recent studies showed that with the technological development of laparoscopic surgery and enhanced recovery programs after surgery (ERAS), elderly CRC could tolerate operation well. The literature suggests that elderly patients benefit from multimodal rehabilitation programs or ERAS in the same way as younger patients (*Van Steenbergen et al., 2013*). Our interaction analysis between the therapy and age in stage III patients showed that the older group benefited more from chemotherapy than younger group (*P*-value = 0.001). And compared with younger patients, older patients were more likely to have fewer lymph nodes sampled and were less likely to receive chemotherapy and radiotherapy. Therefore, current treatment paradigms in the older group may be insufficient. As life expectancy increases, more effective treatments are necessary for the old population.

To reduce bias as much as possible, we used PSM to balance clinicopathological characteristics and used a competing risk model to exclude impact from non-CSD. Finally, we confirmed that elderly CRC was related with more CSD and non-CSD. Age (old) is an independent factor to predict increased CSD. And the Cox model as a sensitivity analysis also had a similar result. Our analysis provides more evidence for elderly CRC receiving stronger adjuvant chemotherapy. The above conclusions can only be acquired from real world data analysis rather than from clinical trials alone.

As a retrospective study, it is impossible to avoid all bias for patient selection. There are several limitations inherent to the database used in the current study. The Charlson index is not available in SEER data. Though the SEER-Medicare can retrieve Charlson index, only patients older than 60 years are registered in their dataset. The Charlson index is strongly related with non-CSD in the old population. Competing risk models can effectively eliminate the impact from unavailable Charlson index. BMI is also an important influence factor in the CRC specific mortality (*Kroenke et al., 2016*). However, the information about BMI in the SEER data is not available. Furthermore, insurance status is an independent risk factor both for advanced disease in their cancer diagnosis and for cancer mortality (*Rosenberg et al., 2015*), and uninsured patients more often have higher T, N, and M stage (*Amini et al., 2016*). But the insurance recode variable is only available from 2007 in the SEER database. Additionally, comorbidities and detailed information about driver gene mutations (KRAS or BRAF) is not available. The variable chemotherapy is only classified as chemotherapy "yes" or "no/unknown" since SEER treatment information cannot accurately distinguish between "no treatment" and "unknown" (*Noone et al., 2016*). Furthermore, the sensitivity of SEER chemotherapy data is only 72.1% (*Noone et al., 2016*), as the detailed regime and duration of chemotherapy is not available in the SEER dataset.

## CONCLUSIONS

In summary, 70 years is a suitable cutoff age to define elderly CRC. Elderly CRC is associated with not only increased non-CSD but also increased CSD. This SEER-based

analysis provides further evidence that current chemotherapy in the elderly may be insufficient. Additional research is required to investigate whether elderly CRC will receive stronger treatment if possible.

### Funding

This work was supported by the Public Welfare Technology Research Program of Zhejiang Province (LGF18H160029), the Jinhua Science and Technology Project (2016-3-005), the Key Program of Jinhua Municipal Science & Technology Bureau (2018-3-001d), and the Key Program of Scientific Research of Jinhua Central Hospital (JY2016-1-02). The funders had no role in study design, data collection and analysis, decision to publish, or preparation of the manuscript.

### Grant Disclosures

The following grant information was disclosed by the authors:
Public Welfare Technology Research Program of Zhejiang Province: LGF18H160029.
Jinhua Science and Technology Project: 2016-3-005.
Key Program of Jinhua Municipal Science & Technology Bureau: 2018-3-001d.
Key Program of Scientific Research of Jinhua Central Hospital: JY2016-1-02.

### Competing Interests

The authors declare that they have no competing interests.

### Author Contributions

- Jianfei Fu conceived and designed the experiments, performed the experiments, analyzed the data, contributed reagents/materials/analysis tools, prepared figures and/or tables, authored or reviewed drafts of the paper, approved the final draft.
- Hang Ruan conceived and designed the experiments, performed the experiments, analyzed the data, contributed reagents/materials/analysis tools, prepared figures and/or tables, authored or reviewed drafts of the paper, approved the final draft.
- Hongjuan Zheng performed the experiments, authored or reviewed drafts of the paper, approved the final draft.
- Cheng Cai performed the experiments, authored or reviewed drafts of the paper, approved the final draft.
- Shishi Zhou performed the experiments, authored or reviewed drafts of the paper, approved the final draft.
- Qinghua Wang performed the experiments, authored or reviewed drafts of the paper, approved the final draft.
- Wenbin Chen performed the experiments, authored or reviewed drafts of the paper, approved the final draft.
- Wei Fu performed the experiments, analyzed the data, prepared figures and/or tables, authored or reviewed drafts of the paper, approved the final draft.

- Jinlin Du conceived and designed the experiments, performed the experiments, authored or reviewed drafts of the paper, approved the final draft.

## Data Availability
Raw data are available in the Supplemental Materials.

## Supplemental Information
Supplemental information for this article can be found online at http://dx.doi.org/10.7717/peerj.6350#supplemental-information.

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
