# Peer review of "Impact of old age on resectable colorectal cancer outcomes"

_PeerJ, doi:10.7717/peerj.6350_

## Round 0.1 · original submission · Major Revisions

I am not sure the cutoff value of 70 was supported by Fig 1, in which the turning point in my view would be around 60. Further, in my understanding both exploratory and validation sets were SEER data. The so-called validation set therefore is not scientifically sound. In addition, there are some grammatical errors. Finally, please provide a point-to-point response to all reviewer comments.

Reviewer 1 ·

Basic reporting

no comment

Experimental design

Line 84, the fourth of exclusion criteria: patients who survived less than one month from diagnosis. This may introduce a selection bias. How many patients were excluded? Why did the researchers exclude these patients?

Validity of the findings

1. Line 286-287, this study concerns the prognostic risk factors of CRC. From the multivariate analysis, researchers concluded that age was an independent risk factor. However, this does not support the conclusion that current therapy standards in the elderly are insufficient because therapy is only a covariate, among many, for survival.
2. Table 3, did all of the patients receive surgery? If not, should surgery be included as a covariate?
3. Overall fitting outcomes (pseudo R2) of the models should be compared, as well.
4. Line 148, what are the methods of univariate and multivariate analysis? Logistic regression?
5. Line 163, for how long did 14,425 and 6,982 patients die of CSD and non-CSD?
6. Line 194-202, researchers concluded 70 years should be adopted as a suitable cut off age. Why was 70 years better as a cut off age, compared with other previously published studies?
7. Figure 2, what does cumulative incidence mean? Overall survival?

·

Basic reporting

The article is very clear with well done background provided.

Experimental design

no comments

Validity of the findings

no comments

Additional comments

Please check the reviewed document attached for additional comments.

Important:- not included BMI and comorbid conditions and substantiate this under study discussion section.

Reviewer 3 ·

Basic reporting

The overall structure of the manuscript is good. However, the method is questionable and the policy implication is not quite sound.

Minor formatting issues:
a. Suggest adding a sample flow chart to show how many patients were excluded by each exclusion criteria
b. Line 92: it should be “(separated, widowed, and divorced)“?
c. Line 127: please keep the decimal consistent throughout the context (i.e. “range 1.0-119.0 months”)
d. Line 196: for the other previously published studies, please add citations
e. Table 1: suggest adding a row to clarity which part is before PSM and which part is after PSM
f. Table 2 and Table 3: please add total sample size, or clarify in the title that Table 3 is after PSM.

Experimental design

1. The objective of this study is stated as “to identify a reasonable cutoff age for defining older patients with colorectal cancer (CRC) and to examine whether old age was related with poor colorectal cancer-specific survival (CSS) and increased colorectal cancer-specific death (CSD)”.
The cutoff age is chosen based on CSS and so age will definitely be associated with CSS. You may want to point out to examine independent effect of older age after controlling for other characteristics.

2. In Figure 1, the turning point looks considerably earlier than 70 years old. Can the author provide some detailed outputs from the Chow test, especially for age 60-70 years old.

3. The statistical analysis part is a little confusing. The author mentioned that the primary end point is cancer-specific survival, and the secondary is cancer-specific death and non-cancer-specific death. For COX model, both death (yes and no) and survival time (months/days) are required for the model, so the endpoints are actually the same. I understand that in the first COX model the author censored the non-CSD, whereas in the second model, the author did PSM first and then treated non-CSD as competing risk. But keeping the second model should be sufficient.

4. Even though the author adopted PSM, the matching didn’t seem to work very well. Quite a few characteristics still showed significantly different distributions across younger and older groups. Ideally, the characteristics should be well balanced between two groups that you only need to run COX model between Age group and CSD after matching.

5. Has the author considered adding insurance status to the model? Medicare may be highly correlated with your age measurement. But can try an “insured vs. uninsured” indicator or "public vs. private vs. none".

Validity of the findings

Age is widely known as one of the most important predictor of survival/death. Especially for older age patients, whose health status sometimes decreases disproportionally. However, I am not quite sure how much this cutoff age can contribute to current clinical practice. The author suggested stronger treatments to the older patients. However, the adjusted model showed that chemotherapy is actually harmful to the patients. (HR of CSD (receiving vs. not receiving chemo =0.84 (0.80-0.88)) I also suggest adding an interaction term between age group and CT/RT to see whether CT/RT affects younger and older patients differently.

---

## Round 0.2 · Minor Revisions

As recommended by one of the reviewers, please consider:

Although the COX regression (to model cause-specific death) and the Fine & Gray competing risk regression are two different models, competing risk model is usually preferred. Also given the similar results using two approaches, the first one (COX model) can be just described as a sensitivity analysis in the context.

Reviewer 1 ·

Basic reporting

No comment

Experimental design

The fourth of exclusion criteria: patients who survived less than one month from diagnosis. This may introduce a selection bias. How many patients were excluded? Why did the researchers exclude these patients?
It has been addressed.

Validity of the findings

1.this study concerns the prognostic risk factors of CRC. From the multivariate analysis, researchers concluded that age was an independent risk factor. However, this does not support the conclusion that current therapy standards in the elderly are insufficient because therapy is only a covariate, among many, for survival.
It has been addressed.
2.Table 3, did all of the patients receive surgery? If not, should surgery be included as a covariate?
It has been addressed.
3. Overall fitting outcomes (pseudo R2) of the models should be compared, as well.
It has been addressed.
4. what are the methods of univariate and multivariate analysis? Logistic regression?
It has been addressed.
5. For how long did 14,425 and 6,982 patients die of CSD and non-CSD?
It has been addressed.
6.researchers concluded 70 years should be adopted as a suitable cut off age. Why was 70 years better as a cut off age, compared with other previously published studies?
It has been addressed.
7.Figure 2, what does cumulative incidence mean? Overall survival?
It has been addressed.

·

Basic reporting

Sufficiently addressed

Experimental design

Rigorous investigation performed

Validity of the findings

no comments

Additional comments

Study is meaningful and useful for further investigation

Reviewer 3 ·

Basic reporting

The author addressed all related comments.

Experimental design

Method:
Please add what models you used for your subgroup analyses (e.g. COX model).

Result:
PSM: balance test of covariates is important after PS matching. A poor prediction model can result in unbalanced covariates after matching. I wonder if the author looked into the model performance (C index). Prior studies have recommended adding interaction terms to improve the model performance.
https://www.ncbi.nlm.nih.gov/pmc/articles/PMC3144483/

If the PS matching does not work and the author still needs to control for the covariates in the final model, why not just run a multivariate regression with the unmatched sample?

Validity of the findings

Data interpretation and conclusion

In table 2 and 3, the reference group of chemotherapy (CT) is "Yes" and the HR is 0.84 (0.80-0.88), which should mean patients not receiving chemotherapy had lower risks of death than patient receiving chemotherapy. In other words, receiving chemotherapy is harmful. In the supplemental figure 2 where the author added age interaction, the coefficients are also <1. Why is the conclusion then "favorite chemotherapy"? Please clarify your reference group in the comparisons.

---

## Round 0.3 · accepted · Accept

All of the reviewer comments are satisfactorily addressed. Thank you for your patience and interest in the PeerJ!